# A comprehensive assessment of multi-system responses to a renal inoculation of uropathogenic *E. coli* in swine

Mohamad Hakam Tiba[1,2]*, Brendan M. McCracken[1,2], Robert P. Dickson[2,3], Jean A. Nemzek[2,4,5], Carmen I. Colmenero[1,2], Danielle C. Leander[1,2], Thomas L. Flott[6], Rodney C. Daniels[2,7,8], Kristine E. Konopka[2,5], J. Scott VanEpps[1,2,8,9], Kathleen A. Stringer[2,3,6], Kevin R. Ward[1,2,8]

1 Department of Emergency Medicine, University of Michigan, Ann Arbor, Michigan, United States of America, 2 Michigan Center for Integrative Research in Critical Care, University of Michigan, Ann Arbor, Michigan, United States of America, 3 Department of Internal Medicine, Division of Pulmonary and Critical Care Medicine, University of Michigan, Ann Arbor, Michigan, United States of America, 4 Unit of Laboratory Animal Medicine, University of Michigan, Ann Arbor, Michigan, United States of America, 5 Department of Pathology, University of Michigan, Ann Arbor, Michigan, United States of America, 6 Department of Clinical Pharmacy, College of Pharmacy, University of Michigan, Ann Arbor, Michigan, United States of America, 7 Department of Pediatrics, Pediatric Critical Care Medicine, University of Michigan, Ann Arbor, Michigan, United States of America, 8 Department of Biomedical Engineering, University of Michigan, Ann Arbor, Michigan, United States of America, 9 Biointerfaces Institute, University of Michigan, Ann Arbor, Michigan, United States of America

* tibam@med.umich.edu

**Data Availability Statement:** All relevant data are within the manuscript and its Supporting Information files. The entire data set can be found

## Abstract

### Background

The systemic responses to infection and its progression to sepsis remains poorly understood. Progress in the field has been stifled by the shortcomings of experimental models which include poor replication of the human condition. To address these challenges, we developed and piloted a novel large animal model of severe infection that is capable of generating multi-system clinically relevant data.

### Methods

Male swine (n = 5) were anesthetized, mechanically ventilated, and surgically instrumented for continuous hemodynamic monitoring and serial blood sampling. Animals were inoculated with uropathogenic *E. coli* by direct injection into the renal parenchyma and were maintained until *a priori* endpoints were met. The natural history of the infection was studied. Animals were not resuscitated. Multi-system data were collected hourly to 6 hours; all animals were euthanized at predetermined physiologic endpoints.

### Results

Core body temperature progressively increased from mean (SD) 37.9(0.8)°C at baseline to 43.0(1.2)°C at experiment termination (p = 0.006). Mean arterial pressure did not begin to decline until 6h post inoculation, dropping from 86(9) mmHg at baseline to 28(5) mmHg (p =

on the NIH Metabolomics Workbench (https://www.metabolomicsworkbench.org/) under the accession # PR000953.

**Funding:** This work is supported in full by the Michigan Center for Integrative Research in Critical Care (https://mcircc.umich.edu/); K.A. Stringer's contribution was supported, in part, by grants from the National Institute of General Medical Sciences (NIGMS; R01GM111400 and R35GM13631201; https://www.nigms.nih.gov/); J. VanEpps' effort was supported, in part, by a grant from the National Institute of Allergy and Infectious Diseases (NIAID; K08AI128006; https://www.niaid.nih.gov/); J. Nemzek's effort was supported, in part, by a grant from the NIGMS (R01GM112799). The content is solely the responsibility of the authors and does not necessarily represent the official views of the NIGMS, NIAID, or the National Institutes of Health. With the exception of MCIRCC, none of the sponsors played any role in the study design. None of the sponsors played any role in data collection and analysis, decision to publish or the preparation of the manuscript. The authors declare no conflict of interest.

**Competing interests:** The authors have declared that no competing interests exist.

0.005) at termination. Blood glucose progressively declined but lactate levels did not elevate until the last hours of the experiment. There were also temporal changes in whole blood concentrations of a number of metabolites including increases in the catecholamine precursors, tyrosine (p = 0.005) and phenylalanine (p = 0.005). Lung, liver, and kidney function parameters worsened as infection progressed and at study termination there was histopathological evidence of injury in these end-organs.

## Conclusion

We demonstrate a versatile, multi-system, longitudinal, swine model of infection that could be used to further our understanding of the mechanisms that underlie infection-induced multi-organ dysfunction and failure, optimize resuscitation protocols and test therapeutic interventions. Such a model could improve translation of findings from the bench to the bedside, circumventing a significant obstacle in sepsis research.

## Introduction

The immune system's dysregulated response to infection (sepsis) remains a major public health threat. The annual incidence rate is over 48.9 million cases globally with 1.7 million in the U.S.A, and a global sepsis-related death rate of 11 million by conservative estimates [1–3]. Notably, the number of reported cases continues to increase [4, 5]. Despite the enormous impact on human health, sepsis, as a disease remains ill-defined with little progress made to further understand its pathogenesis or find effective diagnostics, prognostics or pharmacotherapy regimens. One of the major obstacles to advancing our knowledge of this disease is its unpredictability and the absence of translatable experimental pre-clinical models that accurately recapitulate the evolution of systemic responses to infection in humans [6].

Development of management strategies aimed at altering the inflammatory response due to acute infection starts at understanding the host's response to such triggers, as well as the subsequent sequelae that lead to life threatening organ dysfunction (sepsis) and death. Attempts at decoding such host-dependent changes using pre-clinical and early clinical trials have been repeatedly disappointing and have failed to translate in larger clinical trials [7]. One reason for the decades-long discrepancy between pre-clinical and clinical trials might be the persistent reliance on small animal (especially murine) models in an attempt to identify the precise mechanisms of the systemic inflammatory response. Although rodent-based studies have led to promising mechanistic findings, rodents are phylogenetically distant and biologically divergent from humans, limiting their relevance to the study of certain human diseases such as severe infection and sepsis [8–13]. In addition, these models rarely simulate a natural time-course where comprehensive data are collected from insult to the development of clinically relevant symptoms, and they do not permit easy investigation of the impact of standard therapies such as intravenous fluids, mechanical ventilation, or vasopressor administration over clinically relevant time periods.

Novel pre-clinical models of infection that allow for proper latency and natural progression of the broad range of systemic responses as well as the incorporation of traditional supportive care are more likely to lead to high-fidelity models that mimic the human conditions of infection, sepsis and organ dysfunction. To this end, and consistent with the recently published Minimum Quality Threshold in Pre-Clinical Sepsis Studies (MQTiPSS) recommendations,

more routine use of large animal models for the study of sepsis is emerging [14]. However, current attempts using sheep and swine to model sepsis have mainly utilized endotoxemia [15, 16], peritonitis [17, 18], or direct bacterial IV inoculation [19]. Although these investigations produced valuable insight, they may not be particularly clinically applicable and may not mirror the development and progression of the disease over time.

In this investigation, we present a pilot study aimed at creating a swine model of untreated infection and systemic inflammation that demonstrates the physiologic and immunologic features associated with development of human sepsis using the genitourinary system as the infection source. Such an approach is envisioned to lead to the development of more clinically relevant and reproducible models that will permit mechanistic interrogation and assessment of responses to a variety of treatment scenarios.

## Materials and methods

### Ethics statement

This study adhered to the principles of the National Institutes of Health's *Guide for the Care and Use of Laboratory Animals* [20]. The protocol was approved by the Institutional Animal Care and Use Committee of the University of Michigan (PRO00008551). All animals were procured from the Swine Teaching and Research Center, Michigan State University, East Lansing, MI. All research personnel in this research study received specialized training in animal care and use, swine handling, and surgery from the University of Michigan Unit for Laboratory Animal Medicine. All experiments were terminal.

### Animal instrumentation

Five, male, Yorkshire mix swine weighing a mean (SD) of 45.0 (2) kg were fasted overnight with *ad libitum* access to water. Anesthesia was induced with an intramuscular injection of ketamine/xylazine (20/2 mg/kg) and maintained with 1–2% inhalant isoflurane balanced in room air for the duration of surgical instrumentation. Animals were intubated using a 7.5mm cuffed endotracheal tube and mechanically ventilated with a tidal volume of 7–8 ml/kg. Respiratory rate was adjusted as necessary to maintain a baseline end-tidal $CO_2$ ($PetCO_2$) level between 35-45mmHg (Biopac Systems Inc., Goleta, CA). Core body temperature was maintained between 37.5 and 39.0˚C during instrumentation using a closed loop feedback temperature blanket (Blanketrol, Sub-Zero Medical, Cincinnati, OH). The ECG was monitored continuously using a standard 3-lead configuration (Biopac Systems Inc., Goleta, CA).

Under aseptic conditions, the common carotid artery was cannulated to measure mean arterial blood pressure (MAP) and collect blood samples. The internal jugular vein was cannulated for administration of intravenous anesthetics. The external jugular vein was isolated and cannulated using a pulmonary artery thermodilution catheter (CCOmbo V, 8F, Edwards Lifesciences, Irvine, CA) for measurements of pulmonary artery pressure (PAP), central venous pressure (CVP), and blood sampling. A midline, ~25 cm laparotomy was performed to access the bladder for placement of a Foley catheter. A small retroperitoneal window was opened to reach the kidney and the ureter. Surgical procedures were carried out concurrently within 1.5 to 2 hours.

At the end of surgical instrumentation, animals were transitioned to total intravenous anesthesia using midazolam (5–20 mcg/kg/min), fentanyl (0.1–0.5 mcg/kg/min), and propofol (10–1000 mcg/kg/min) to mitigate the effects of isoflurane on the cardiovascular system during the prolonged period of observation post inoculation. The level of anesthesia was monitored by assessing corneal reflex, jaw tension, and hemodynamics including blood pressure, heart rate and respiratory rate.

## Inoculation

The clinical, uropathogenic isolate (CFT073) of *Escherichia coli* [21] from a cryopreserved glycerol stock (-80˚C) was streaked onto Luria Bertani (LB) agar plates. Single colony inoculates were grown to stationary phase overnight (shaken at 200rpm at 37˚C) in 200ml of Luria Bertani broth. The cells were pelleted by centrifugation (4000rpm, 6min), washed twice with 1x phosphate buffered saline (PBS), and resuspended in 6ml 1xPBS. A 22G catheter was inserted into the lower-most part of the kidney under direct visualization through the laparotomy. Inoculation was achieved by an infusion of 5ml of bacterial solution directly into the renal parenchyma over 15 minutes. This resulted in an average (SD) total inoculum per animal of $3.1x10^{11}$ ($2.5x10^{10}$) cells, quantified by incyto chip hemocytometer. The laparotomy was immediately closed following the inoculum infusion. The corresponding ureter was occluded (with an externalized vessel loop) for one hour then released.

## Monitoring and sample collection

In order to observe the natural course of the infection, animals were observed with limited intravenous supportive care (30 mL/hour of normal saline) for the duration of the experiment, up to 24 hours. Hemodynamics and ventilation parameters such as MAP, PAP, CVP, heart rate (HR), core body temperature, $PetCO_2$ and fraction of oxygen in the inspired and expired air ($FiO_2$, $FeO_2$) were monitored and recorded every hour using a Biopac Data Acquisition System MP150 (Biopac Systems Inc., Goleta, CA). Arterial and mixed-venous blood gases were obtained every two hours and analyzed with temperature correction using ABL800 FLEX (Radiometer America, Brea, CA). Blood samples for complete blood count (CBC), comprehensive chemistry profiles, (Vetscan HM5 & VS2, Abaxis, Union City, CA) as well as for the cytokines, tumor necrosis factor (TNF)-$\alpha$ and interleukin (IL)-6 (Porcine TNF$\alpha$ DuoSet ELISA & Porcine IL-6 DuoSet ELISA, R&D Systems, Minneapolis, MN) were obtained and analyzed every 6 hours. In one animal, blood samples to measure TNF-$\alpha$ and IL-6 were collected every hour for the first 6 hours, and then every 6 hours to check for the acute phase response. Every 6 hours arterial and mixed venous blood samples were also collected for the measurement of metabolites by quantitative $^1$H-nuclear magnetic resonance (NMR) spectroscopy (see S1–S4 Figs, S1 and S2 Files, S1 and S2 Tables).

The experiment was terminated and animals were euthanized using 1–2 mEq/kg of potassium chloride while under anesthesia if one of the following humane endpoint conditions was met: persistently low MAP ($< 40$ mmHg for 2 hours) or low $PetCO_2$ ($< 25$ mmHg for 2 hours) or 24 hours of total experiment duration. Upon completion of the study, tissue samples from the lungs (*en bloc*), liver, spleen, intestine, and kidneys were collected, placed in formalin, processed for pathology examination or stored for future analysis. Whole blood and plasma samples were also collected and stored for future analysis.

## Statistical analysis

Descriptive data are presented as the mean and standard deviation (SD). Statistical analyses were performed using repeated measures analysis of variance (ANOVA) with post-*hoc* correction of multiple comparisons using Tukey's test as applicable. In cases of missing data, a mixed-effects analysis was used. Whole blood metabolomics data were analyzed as described in the S1–S4 Figs, S1 and S2 Files, S1 and S2 Tables. In all cases, significance was considered at $\alpha = 0.05$. Data were analyzed using MATLAB R2017a (The MathWorks, Inc., Natick, MA) and PRISM 8 (GraphPad Software, San Diego, CA).

## Results

All animals used in this pilot study died or reached *a priori* study end points with a mean (SD) survival time of 11.6(1.5) hours with a survival time range between 9 and 13 hours. S1 Table in the supporting information file lists baseline and end of experiment hemodynamic, clinical chemistry, and oxygenation data.

### Hemodynamic and physiologic parameters

There were noted changes in a number of hemodynamic and physiologic parameters as systemic response to infection progressed. Mean core body temperature increased by 13.5% (37.9 (0.8) to 43.0(1.2) $^{\circ}$C, p = 0.006; (Fig 1A) from baseline to the end of the experiments. This occurred rapidly at an average rate of 0.43 $^{\circ}$C/h. Heart rate (Fig 1B) progressively increased from 74.2(6.0) at baseline to 142.0(43.2) beats per minute (BPM) at the termination of the experiment but this did not reach statistical significance. MAP decreased from 86(9) mmHg at baseline to a low of 28(5) mmHg (p = 0.005) at the end of the experimental time period (Fig 1C). The central venous pressure (Fig 1D) also followed a downward trend; it was lower at baseline than at the termination of the experiment (7.7(1.0) mmHg vs 4.7(0.48) mmHg;

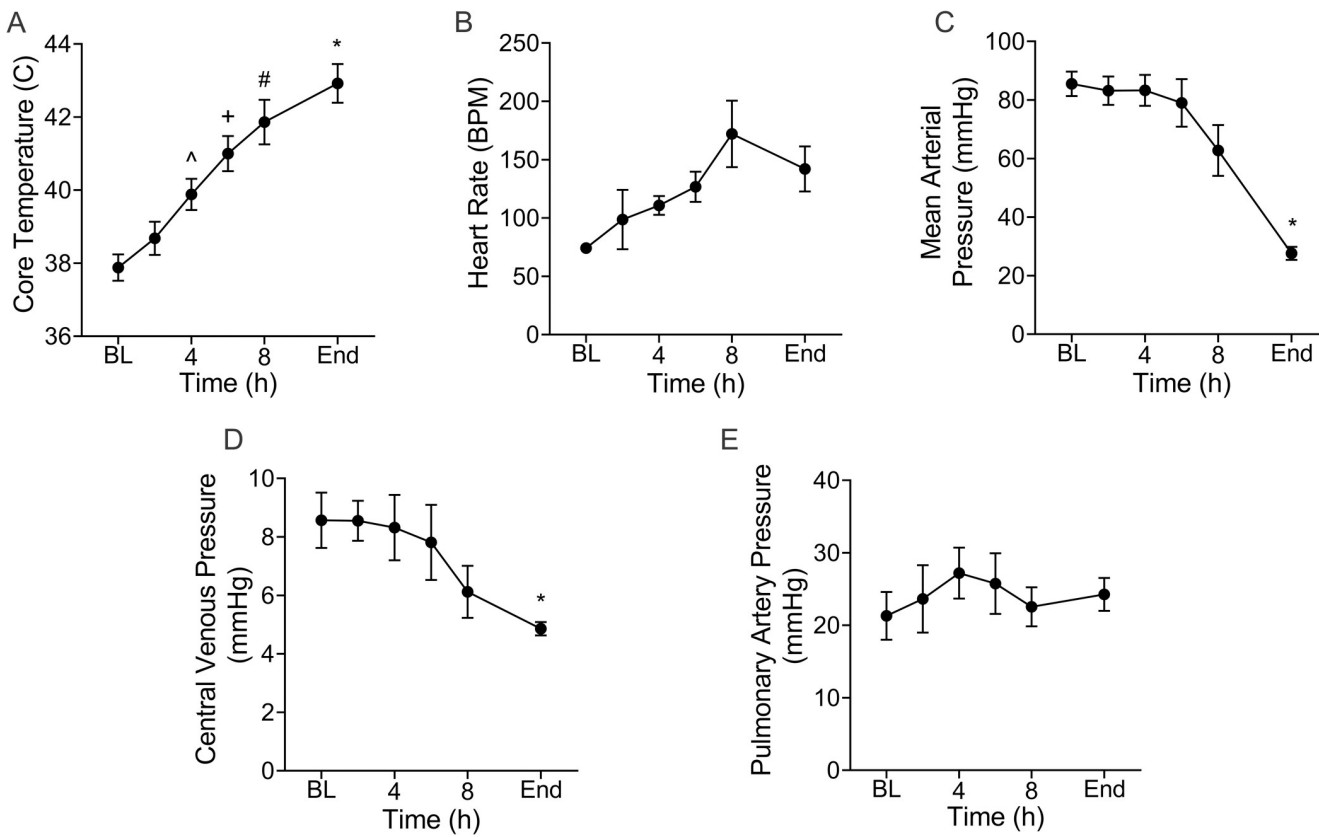

**Fig 1. Infection leads to progressive changes in hemodynamic and physiologic parameters.** From baseline (BL), core temperature (**A**) was increased at 4h ($^{\wedge}$p = 0.03), 6h ($^{+}$p = 0.01), 8h ($^{\#}$p = 0.01) and at the end (*p = 0.006) of the experiments. Heart rate (**B**) also increased but the changes were not significant. Mean arterial pressure (**C**) progressively declined and was significantly lower at the end of the experiments compared to BL (*p = 0.005). There was a progressive decline in central venous pressure such that by the end of the experiments it was lower than BL (*p = 0.05) (**D**). Pulmonary artery pressure was highly variable but trended upward in the early phase of the experiment likely reflecting a compensatory increase in cardiac output driven by an increase in heart rate. Data are the mean(SE) of five animals. $^{\wedge}$ Denotes significant difference between baseline and 4h. $^{+}$ Denotes significant difference between baseline and 6h. $^{\#}$ Denotes significant difference between baseline and 8h. * Denotes significant difference between baseline and end of experiment.

(p = 0.03). Conversely, pulmonary artery pressure (Fig 1E) varied between 20–28 mmHg over the course of the experiment. Respiratory rate did not change as these animals were deeply anesthetized and mechanically ventilated.

## Clinical chemistries

A number of organ function tests were measured over the experimental time course. Whole blood creatinine (Fig 2A) and BUN (Fig 2B) levels increased from 12(1.0) to 32(9) µmol/L (p = 0.02) and 2.6(1.2) to 7.6(2.3) mmol/L (p = 0.003), respectively. Glucose (Fig 2C) progressively declined over time from the 2h time point. The mean baseline value of 8.5(1.4) mmol/L was significantly higher than the mean of 3.7(1.2) mmol/L (p = 0.04) at the end of experiment. This is consistent with findings from the metabolomics data described below. Animals also developed higher lactate levels (Fig 2D) at the end of the experiment 8.0(3.8) when compared to baseline 1.5(0.9) mmol/L (p = 0.09). When taken together, the high level of lactate and low MAP (Fig 1C) at end of experiment, are consistent with the development of shock. Metrics of liver function such as total bilirubin (Fig 2E) and alanine aminotransferase (ALT; Fig 2F) did not show any statistically significant changes over the course of the experiment.

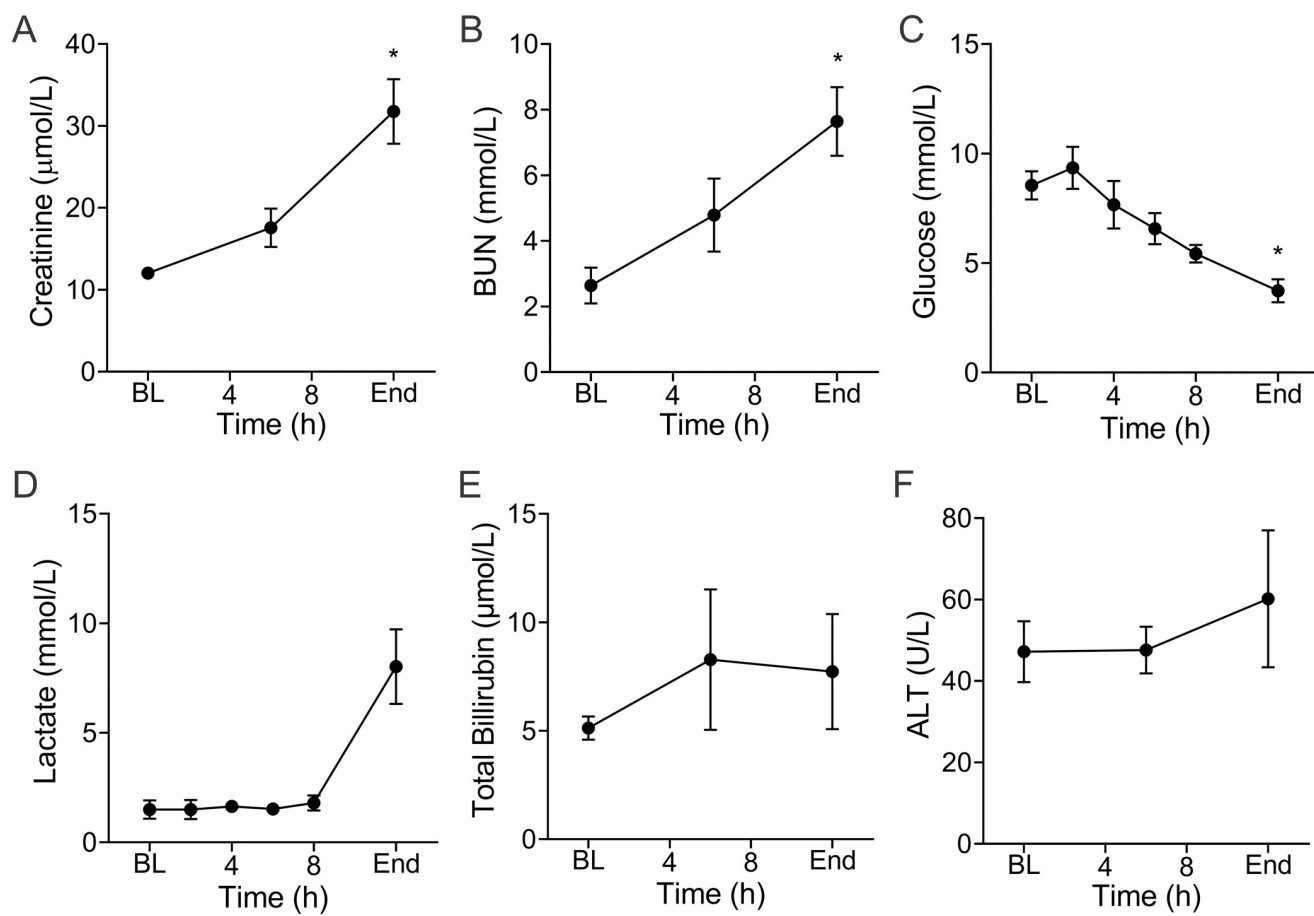

**Fig 2. Infection induces changes in clinical chemistries.** Whole blood creatinine (**A**) progressively increased over the course of the experiment and was significantly higher at the end compared to baseline (BL) (*p = 0.02). Blood urea nitrogen, BUN (**B**), followed the same trend (*p = 0.002). Over the course of the experiments, whole blood glucose (**C**) progressively declined and was significantly lower compared to BL by the end of the experiments (*p = 0.04). Lactate concentration (**D**) remained unchanged until the terminal phase of the experiment when it increased (p = 0.09 vs baseline). Markers of liver function including total bilirubin (TBIL; **E**), alanine transaminase (ALT; **F**) were not significantly changed but trended upward. Data are the mean(SE) of five animals. * Denotes significant difference between baseline and end of experiment.

## Hematologic parameters and white blood cell count and differential

The mean platelet count decreased from 292(114) to 203(102) ($10^9$ cells/L) however, this difference was not significant (Fig 3A). Hemoglobin (Fig 3B) progressively increased from 10.8 (1.6) to 14.6(1.8), which is most likely indicative of hemoconcentration. Hematocrit measurements further support this as they increased from 40.9(6.6) to 53.8(3.6) % (p = 0.05) (Fig 3C).

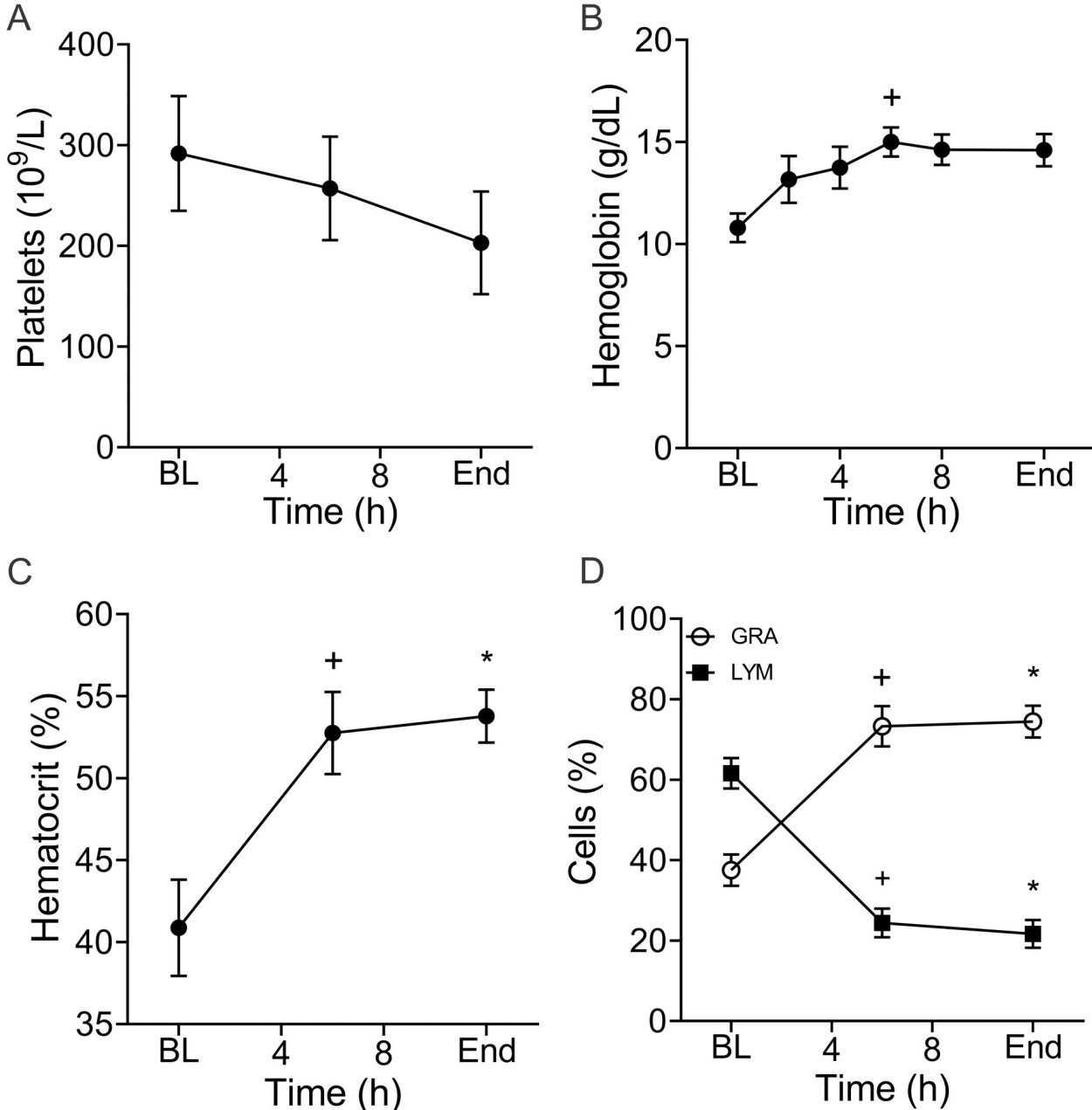

**Fig 3. Hematologic parameters and white blood cell differentials are altered as infection progresses.** Mean platelet count (**A**) declined as infection progressed but these changes were not statistically significant. Over time, hemoglobin (**B**) increased from baseline (BL; 6h: $^+$p = 0.005). In addition, there was a profound shift in the white blood cell (WBC; $10^9$/L) differential (**C**) in which the percent (%) of neutrophils (GRA) increased from BL (6h: $^+$p = 0.02; End: $^*$p = 0.01) and lymphocytes (LYM) declined (6h: $^+$p = 0.01; End: $^*$p = 0.004). Data are the mean(SE) of five animals. $^+$ Denotes significant difference between baseline and 6h. $^*$ Denotes significant difference between baseline and end of experiment.

Mean white blood cell count (WBC) increased from 17.0(3.6) to 18.4(8.1) $10^9$/L, but this was not significant (Fig 3D). However, there was a significant dynamic shift in WBC differential for which neutrophils increased from 37.5(8.7) % to 74.4(8.9) % (p = 0.01) and lymphocytes decreased from 61.6(8.5) % to 21.6(7.7) % (p = 0.004).

## pH, blood gases and oxygenation

The arterial pH and $HCO_3^-$ declined over time (Fig 4A and 4B), and the partial end tidal of carbon dioxide ($PetCO_2$) remained stable indicating development of a metabolic acidosis (Fig 4C). Overall, oxygen consumption (Fig 4D) remained stable but began a declining trend during the latter hours of the experiment. Notably, the arterial oxygen partial pressure ($PaO_2$) to fractional inspired oxygen ($FiO_2$) ratio progressively declined over time (Fig 4E) from 457(62) at baseline to 315(118) at experiment termination. This did not reach statistical significance (p = 0.20) likely due to the high variability at the terminal phase of the experiment. However,

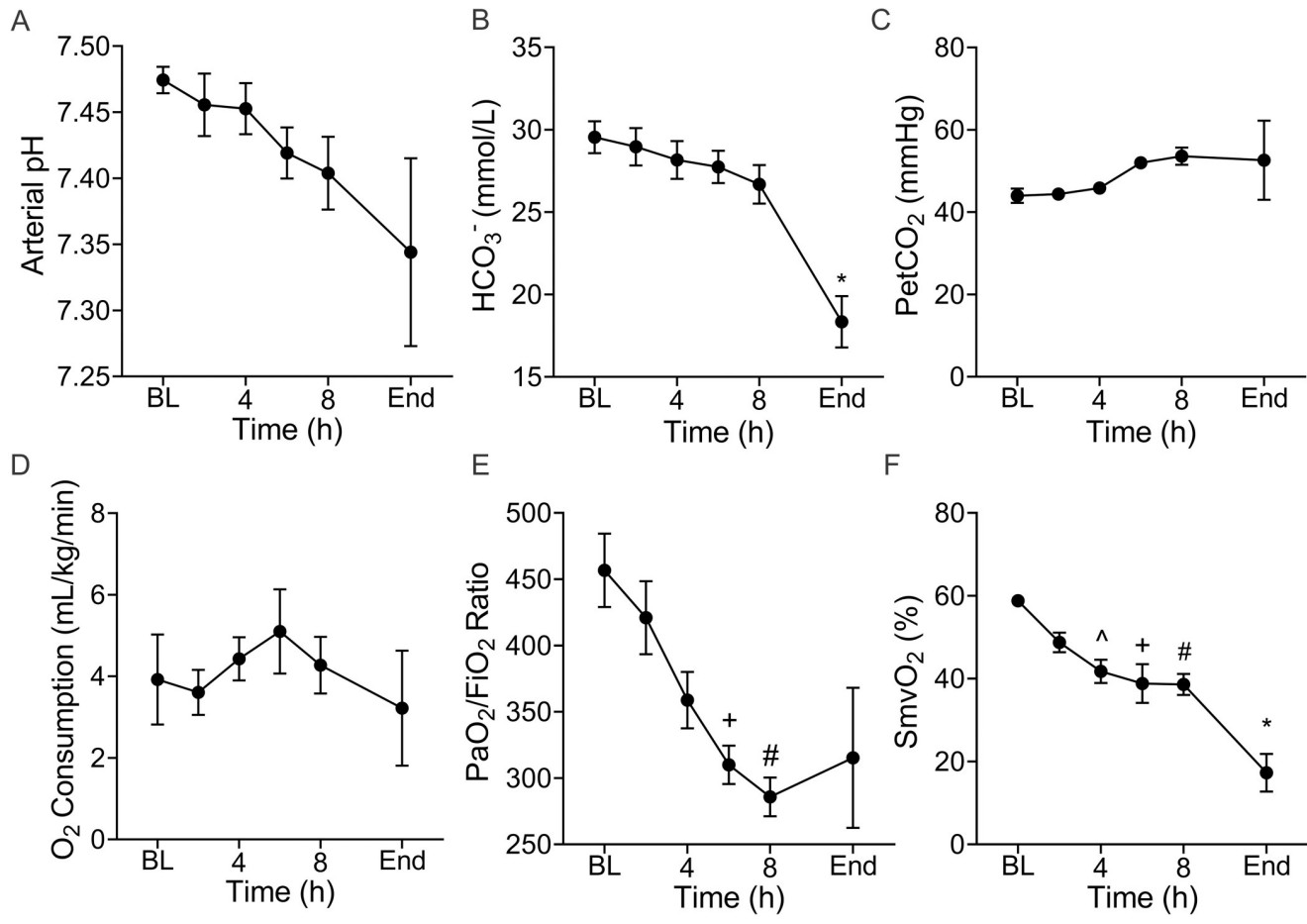

**Fig 4. Infection disrupts acid-base and oxygenation homeostasis.** Arterial pH (**A**) progressively declined from baseline (BL) of the end of the experiments but these changes did not reach statistical significance. Similarly, bicarbonate ($HCO_3^-$) concentration declined over time (**B**) and the end value was significantly different from BL (*p = 0.02). In accordance with the mechanical ventilation protocol, the (**C**) positive expiratory pressure (Pet) of $CO_2$ remained relatively constant. Oxygen consumption (**D**) was highly variable but trended downward toward the end of the experiments. However, the oxygen partial pressure ($PaO_2$) to fractional inspired oxygen ($FiO_2$) ratio (**E**) continuously declined from BL to 6h (+p = 0.005) and 8h (#p = 0.001). Mixed venous oxygen saturation ($SmvO_2$) progressively declined (F) from BL to 4h (^p = 0.009), 6h (+p = 0.05), 8h (#p = 0.01) and the end (*p = 0.005) of the experiment. Data are the mean(SE) of five animals. ^ Denotes significant difference between baseline and 4h. + Denotes significant difference between baseline and 6h. # Denotes significant difference between baseline and 8h. * Denotes significant difference between baseline and end of experiment.

the difference between the mean $PaO_2/FiO_2$ ratio at baseline and 8h (286(33)) was markedly different (p = 0.01). Mixed venous oxygen saturation (Fig 4F) also declined over time from baseline 59(1.9) % reaching a critically low value of 17(10.2) % at the end of the experiment (p = 0.005).

## Metabolic measurements

Blood samples for the acquisition of metabolomics data were available from 4 of the 5 animals. The concentration of a number of metabolites changed during infection progression. Metabolites with a false discovery rate (FDR)-corrected p value of < 0.10 are shown in Fig 5. The most significantly changed metabolite, as determined by FDR-corrected ANOVA p value (0.05), was creatine (Fig 5A). Notably, phenylalanine (Fig 5B) and tyrosine (Fig 5C), precursors of catecholamines, progressively increased. Broadly, with the exception of glucose (Fig 5G), all measured metabolite concentrations increased over time, including lysine (Fig 5D), histidine (Fig 5F), glutamine (Fig 5H) and methionine (Fig 5I). The concentration of inosine monophosphate (IMP) also progressively increased (Fig 5E). This suggests disruption of adenosine metabolism but the calculated adenylate energy charge did not change although energy balance (depicted by the ATP/ADP ratio) trended lower (see S1A Fig and S1B Fig in the online supporting information). Notably, increases in lactate concentration were modest but greatest towards the end of experimentation which is consistent with the clinical measurements (see S1C Fig in the online supporting information and Fig 2D). All metabolites with an FDR-corrected ANOVA p value of < 0.15 are shown in S1 Fig. The list of all detected metabolites can be found in S2 Table.

## Plasma cytokines

Plasma IL-6 was elevated in all animals, increasing 15-fold from a baseline to 6 hours with a mean(SD) of 394.5(446) to 6339.4(5648) pg/mL respectively. (Fig 6 inset). Mean plasma TNF-α (Fig 6 inset) declined from a baseline of 2286(1259) to 741.3(549) pg/mL at the end of the experiment. The change in both IL-6 and TNF-α were not statistically significant. However, in one animal from which hourly blood samples were collected for 6 hours, the measured TNF-α peak occurred 1-hour post inoculation, 3.36 (0.03) x $10^4$ pg/mL and IL-6 at 2hours post inoculation 4.26 (0.49) x $10^4$ pg/mL (Fig 6). The levels of TNF-α and IL-6 rapidly decreased by hour 6 to 973 pg/mL and 6031 pg/mL, respectively. This suggests that the early peak level of TNF-α and IL-6 were missed by sampling at 6-hour intervals.

## Histopathology of vital organs

Representative sections of the kidneys, liver, and lungs were processed and hematoxylin and eosin-stained slides were made. These tissues showed histomorphologic evidence of organ injury. Significant histological findings in the inoculated kidneys included marked neutrophilic tubulointerstitial nephritis with evidence of vascular origin (Fig 7A and 7B). Inflammatory foci appeared to be preferentially centered on the venous vessels, but the extent of tissue damage, presence of bacteria in renal tubules, hemorrhage, and inflammation obscured the origin of histologic lesions. There were no significant histopathologic lesions observed in examined sections of the contralateral kidneys. The liver showed marked acute sinusoidal dilation (Fig 7C) consistent with acute vascular (agonal) congestion. Sections of the lung showed a patchy inflammatory infiltrate that included mild interstitial thickening by mononuclear inflammatory cells (Fig 7D) and focal acute bronchopneumonia. The acute bronchopneumonia was seen as neutrophils infiltrating the bronchiolar epithelium (Fig 7F), accompanied by a cellular intra-alveolar airspace exudate in which neutrophils predominated (Fig 7G and 7H).

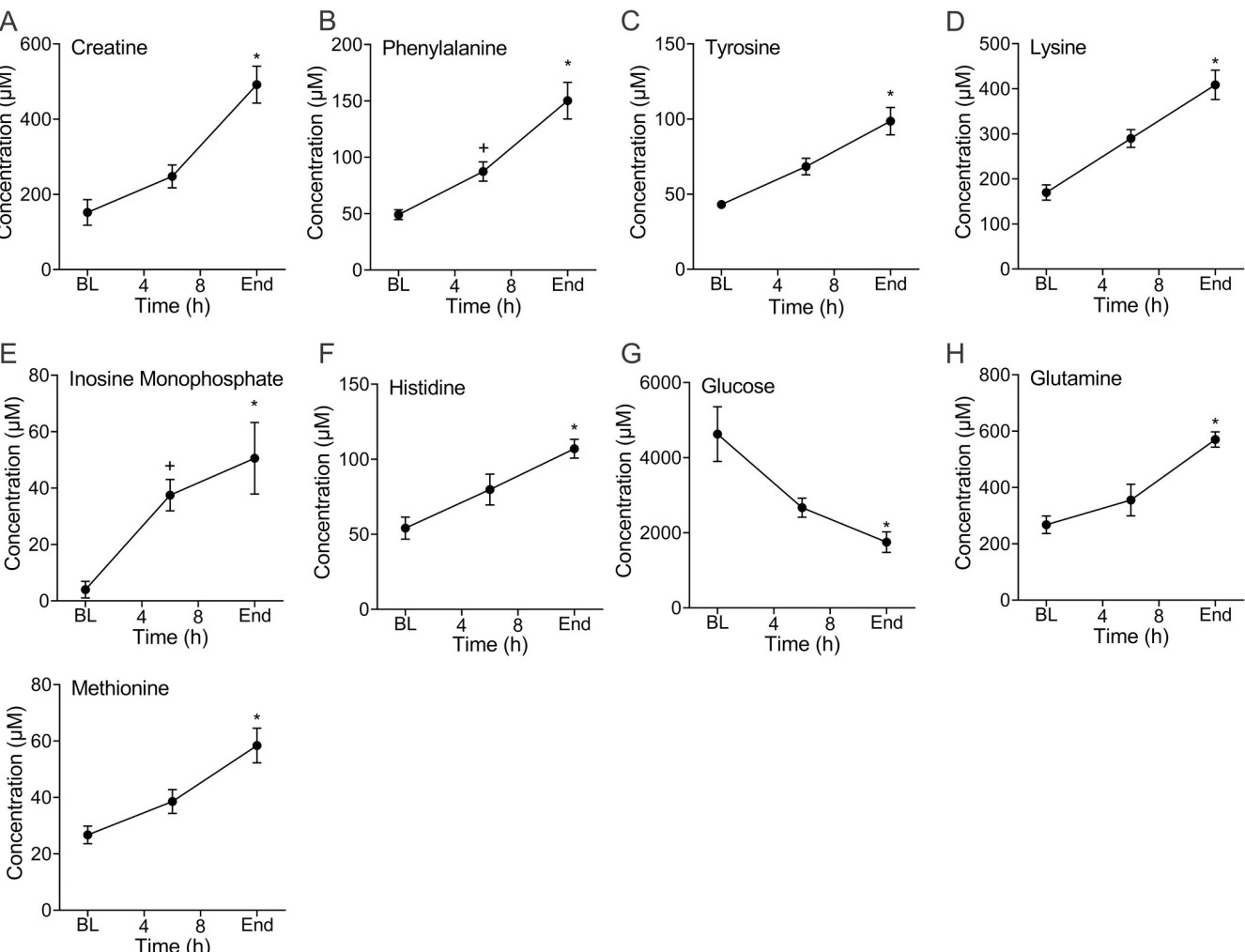

**Fig 5. Infection causes a large shift in host metabolism as measured by whole blood metabolomics.** Creatine (**A**), a non-proteinogenic amino acid which is abundant in skeletal muscle and critical to energy metabolism, increased in concentration from BL the end of the experiments (End: *p = 0.01). The essential amino acid, phenylalanine (**B**), increased from baseline (BL) as systemic infection progressed (6h: +p = 0.02; End: *p = 0.01). In parallel, the non-essential amino acid, tyrosine (**C**), that is produced from phenylalanine, also increased from BL (End: *p = 0.005). Lysine (**D**), an essential amino acid, progressively increased over the course of infection (End: *p = 0.0002). Inosine monophosphate (**E**), which is a nucleotide monophosphate, also progressively increased from BL (6h: *p = 0.006; End: *p = 0.03). The essential amino acid, histidine (**F**), which is a precursor of histamine, was increased compared to BL at the end of the experiment (End: *p = 0.04). Consistent with the clinical measurement, glucose (**G**), progressively declined from BL to the end of the experiment (*p = 0.05). Glutamine (**H**), a non-essential amino acid, and methionine (I), an essential amino acid, both of which are important in maintaining glutathione homeostasis, were higher at the end of the experiments than at BL (End: *p = 0.006 and *p = 0.0001, respectively). Data are the mean(SE) of four animals. Metabolites are presented in ascending rank order of their FDR-corrected ANOVA p value which in all cases was less than 0.10. + Denotes significant difference between baseline and 6h. * Denotes significant difference between baseline and end of experiment.

These findings were affiliated with variably dense perivascular inflammation, comprised mainly of lymphocytes and lymphoid aggregates, and rare fibrin thrombi (Fig 7E) in small blood vessels.

## Discussion

In this investigation, we studied the progression of untreated severe infection initiated by a direct inoculation of a virulent clinical isolate of *E. coli* (CFT073) [22, 23] into the kidney parenchyma of swine. This direct renal inoculation provoked a systemic inflammatory

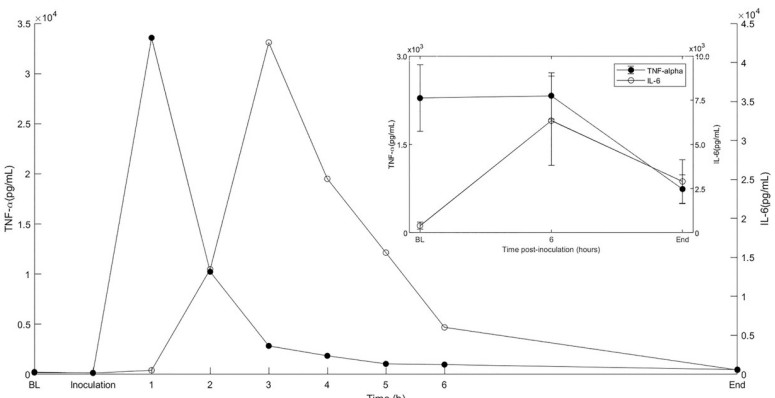

**Fig 6. The inflammatory cytokines, tumor necrosis factor (TNF)-α and interleukin (IL)-6 increase in during infection.** Plasma TNF-α and IL-6 were elevated early in the course of infection (inset). Data are the mean(SE) of five animals. In one of these animals, plasma samples were collected hourly for the first 6h of the experiment. These data show that the measured TNF-α peak concentration occurs very quickly and precedes that of IL-6.

syndrome which led to the development of organ dysfunction and shock. Importantly, the model permitted the acquisition of simultaneous multi-organ system temporal data that enabled the tracking of key events of the progression of localized infection to systemic manifestation (Fig 8). This type of model may circumvent many of the challenges and limitations of current rodent models that have failed to widely translate to the human situation, including cecal ligation and puncture in which bacterial species involved are rarely defined, ongoing intestinal ischemia and necrosis is concurrently present, and the mouse microbiota is not the same as in human patients [24, 25].

There are a number of important hallmarks that we identified in our model of systemic responses to a renal inoculation of uropathogenic *E. coli*. Despite a rapid rise in core body temperature, MAP was maintained for approximately 6 hours after which it began to rapidly decline; MAP is a measurement of the cardiovascular component of the sequential organ failure assessment (SOFA) score which is clinically used to diagnose and assess sepsis severity

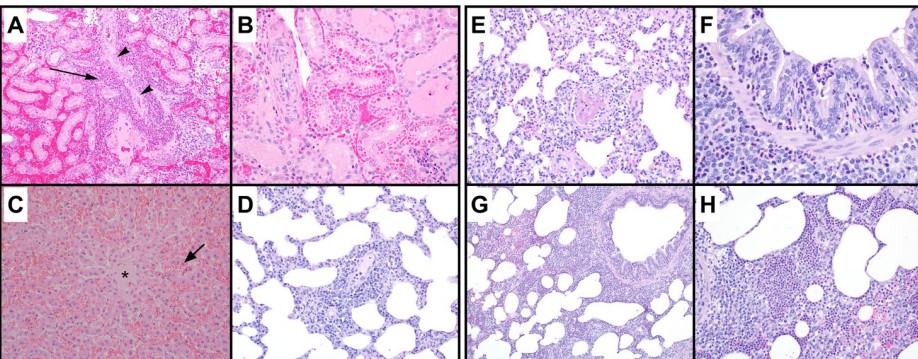

**Fig 7. Representative hematoxylin and eosin stained sections of renal, hepatic, and lung tissues.** (A) foci of suppurative inflammation (arrow) and degenerate cellular debris adjacent to a renal arteriole (arrowhead), (B) focus of renal tubules with intracytoplasmic eosinophilic droplets, and (C) Acute hepatic sinusoidal dilation (arrow) is seen around the central vein (*). Lung tissues show (D) perivascular interstitial inflammation, acute bronchopneumonia with neutrophilic infiltrates (F, G, H) within alveolar airspaces (H), focally in bronchiolar lumens (F), and in peribronchiolar airspaces (G); Rare fibrin thrombi (E) were identified in small precapillary vessels. Renal tissues were collected from the inoculated kidney.

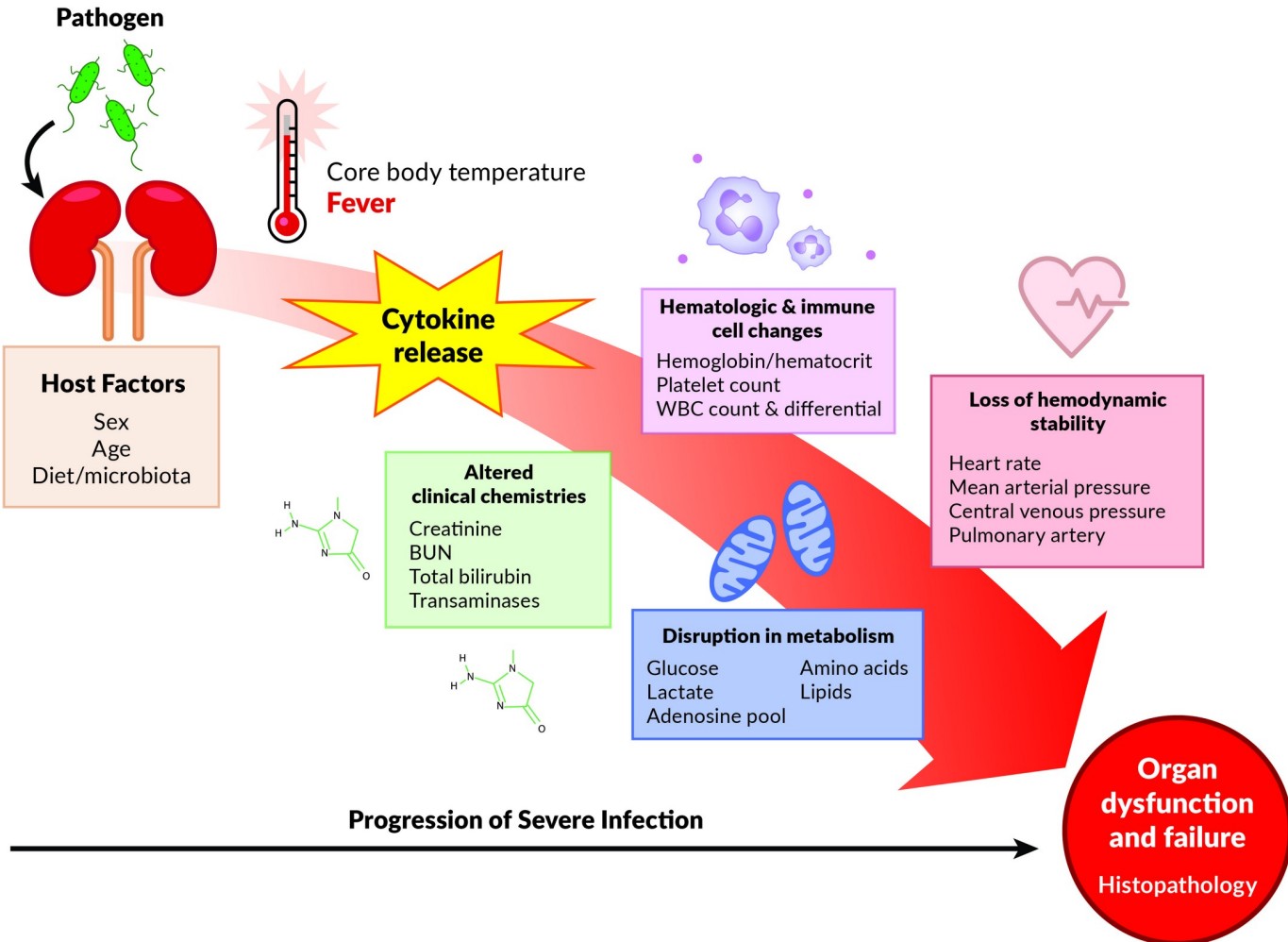

**Fig 8. Experimental modeling of the natural history of systemic responses to infection.** A "high-fidelity" longitudinal swine model of infection induced by injection of pathogenic *E. coli* into the kidney parenchyma leads to numerous events including but not limited to fever, altered clinical chemistries, increases in blood inflammatory cytokines, alterations in hematologic and immune cell numbers, disruption in metabolism and ultimately, loss of hemodynamic stability. These events culminate as end-organ dysfunction and injury. This model permits the measurement of numerous parameters as infection progresses which allows for the detailed study of key features that lead to, and mechanisms that underlie, organ dysfunction and failure. It also allows for the manipulation of host factors such as sex, age and diet. Improved understanding of the temporal relationship of these events will help drive well-informed testing of intervention strategies including fluid resuscitation and targeted therapeutics.

[26]. Heart rate followed a similar trajectory to body temperature. For most infections, heart rate increases by ~8 BPM for every 1°C rise in body temperature [27]; this is evident in our model. In aggregate, these changes and their temporal associations are difficult, if not impossible, to study in humans because most often patients self-administer anti-pyretic medications before presenting to the health care system. Studying fever during the natural history of infection is relevant because it is an important and common response and is generated by a number of mechanisms including increases in the concentrations of the inflammatory cytokines, TNF-α and IL-6 [28]. It also has significant physiologic implications. For example, fever is an important mediator of the host response and if untreated, can contribute to the translocation of gut bacteria and cause cell death and end-organ injury [28].

Changes in other parameters altered by infection were captured by the measurement of clinical chemistries including glucose and lactate. Blood glucose has been an intense focus of

study in critically ill patients including those with sepsis. It is generally accepted that hyperglycemia is a stress response and it has been shown to be associated with illness severity [29]. However, the study of glucose in the clinical setting of sepsis is challenging due to concomitant illnesses like diabetes and variable nutritional support and in some cases, sepsis can present with hypoglycemia [30]. Here we demonstrate that glucose concentration declines during the course of infection as measured by both a clinical assay and metabolomics which may be counterintuitive given the elevation in levels of metabolic precursors to catecholamines. Other clinical chemistries directly related to organ function were also measured. As expected, whole blood creatinine, BUN, total bilirubin increased and these changes occurred in advance of a significant decline in MAP. Creatinine and total bilirubin represent the renal and liver components, respectively, of the SOFA score [26].

Other indications of infection were evident in measurements of platelet count, which declined over time. Acute thrombocytopenia can accompany sepsis and its persistence is associated with mortality [31]. As such, platelet count represents the coagulation component of the SOFA score [26]. In our experiments, the WBC count did not change. However, there was a rapid shift in the WBC differential in which neutrophils increased and lymphocytes declined. This pattern of change is consistent with physiologic stress, indicative of acute illness and often precedes an elevation in WBC count [32, 33]. It has been proposed that this phenomenon is due to an abrupt increase in catecholamines [34].

Arterial pH began to drop well before any change in blood lactate concentration and more closely coincided with an increase in whole blood creatinine. This is consistent with the finding that the metabolic acidosis of infection is attributable to renal dysfunction rather than hyperlactatemia [35]. Despite animals not being paralyzed, a compensatory hyperventilation was not observed likely due to the deep state of anesthesia produced. The progression of infection was also met with a rapid decline in lung function as measured by the $PaO_2/FiO_2$ ratio. By the 8 h time point, animals reached a $PaO_2/FiO_2$ ratio that is indicative of acute hypoxemic respiratory failure. The $PaO_2/FiO_2$ ratio represents the pulmonary component of the SOFA score [26].

Infection induced a number of metabolic changes that extend beyond those which have been traditionally studied (e.g., glucose and lactate). Notably, this included a number of both essential and non-essential amino acids (see S1–S4 Figs, S1 and S2 Files, S1 and S2 Tables) many of which are metabolically related (see S2 Fig in the online supporting information). These findings also highlight the rapid onset of these changes that occurred in advance of changes in lactate concentration, in particular, increases in phenylalanine and tyrosine. Phenylalanine is a precursor of tyrosine and both amino acids are required for the synthesis of catecholamines [36] which are necessary for brain function and to support blood pressure. Although we did not specifically measure catecholamines, this illustrates the utility of our model for the study of infection-induced changes in metabolism and the value of serial sampling that is afforded by a large animal model. We acknowledge that in this work, conclusions about a broad range of metabolic pathways is limited by the chosen analytical platform, in this case, quantitative $^1$H-NMR. This approach primarily detects polar compounds. However, this is not a limitation of the model itself as any assay can be performed to assess components of metabolism including the energy demands induced by infection.

In concert with the pyretic response, TNF-α and IL-6 were elevated in response to infection. In addition to their role in the manifestation of fever, both are associated with severity of organ failure and are predictive of mortality in humans with sepsis [37, 38]. TNF-α increases in the early phases of the inflammatory response [37] then declines in the chronic phase partly due to the inhibitory effect of IL-6 [37]. We acknowledge that our initial sampling scheme was not optimal as we missed the early increases in both cytokines due to our 6-hour sampling

frequency. However, in one animal with more frequent blood measurements (hourly for the first 6 hours), we were able to capture a more granular measurement and time course of both cytokines corroborating the systemic inflammatory response in this model. Of note, as shown in Fig 6, there was an early post-inoculum rise in concentrations of TNF-α and subsequently IL-6, that appears to occur independent of the invasive surgical procedures. This suggests that such increases are primarily due to the host's response to the newly introduced bacteria into the kidney.

Swine have been used in biomedical research for more than 25 years to model human disease. Although phylogenetically distant from humans, unlike rodents, their anatomical, hemodynamic, and physiological similarities to humans [39] make them an ideal species for modeling the systemic responses of infection [40] due to their comparable endotoxin sensitivity and tissue antigenicity to humans. The large size of swine allows for comprehensive surgical instrumentation, adequate longitudinal blood and tissue sampling, and application of therapeutic interventions such as mechanical ventilation. Swine also have an established track record in sepsis and critical care research which includes studies that have shown hemodynamic changes and cytokine profiles that are similar to those of humans [41–44]. We have exploited these advantages and modernized the model by employing a clinically faithful organ specific infectious source (renal injection of *E. coli*) to acquire detailed, real-time hemodynamic, physiologic and biochemical data [14] that, as far as we know, has not been done to this extent in other swine sepsis models (see S1 Table). The kidney was chosen for inoculation due to its small size, ease of access, and the reproducibility of the insult location. Furthermore, there is clinical relevance since urinary tract infections that lead to pyelonephritis are common. Acute pyelonephritis occurs at a rate of 15 to 17 cases per 10,000 females and 3 to 4 cases per 10,000 males annually in the US [45]. It is a common source of gram negative infection and also accounts for 25% of sepsis cases annually [46]. Other conditions such as soft tissue, pulmonary, and GI tract infection are arduous to model in large animals due to the larger size of the organs, complexity of the microbiome, and the time needed for the manifestation of illness. These modifications make our model novel and distinct from other porcine models of sepsis particularly since, to date, most have induced sepsis via peritonitis [17, 18, 40, 42, 43] or by direct intravenous infusion of either a pathogen [19, 44] or endotoxin [41]. In summary, our model is closely aligned with the current MQTiPPS review Part II recommendations which advocate for the discontinuation of endotoxin challenge models as well as "modeling sepsis syndromes that are initiated at sites other than the peritoneal cavity (e.g., lung, urinary tract, brain)" [47].

Importantly, our model has implications for the study of sepsis and septic shock in relation to the components of the definition of sepsis [48], and all but one (neurological) of the components of the SOFA score, can be assessed [26]. Recently, the definition of sepsis has expanded beyond the systemic inflammatory response syndrome (SIRS) to a combination of a dysregulated immune response with development of life threatening organ dysfunction while also providing different criteria for septic shock based on changes in blood pressure and lactate levels [48]. Although, currently there has been a shift towards more emphasis on prognostication using SOFA grading and scores rather than SIRS criteria, we evaluated the systemic responses in these animals based on both doctrines. The late rapid increase in lactate combined with the decrease in blood pressure and increased heart rate raises the possibility that these animals developed systemic sepsis and succumbed to shock.

In this pilot study, we elected to assess the natural evolution of the systemic responses to untreated infection (minimal fluid supplementation and no antibiotics) so that the experimental timeline of disease progression and lethality could be established. This is necessary to direct the design of future planned studies. We envision that this model, and future derivations of it

such as utilizing a full resuscitation protocol with fluids and antibiotics, use of a drug resistant *E. coli* to mitigate the effects of antibiotics, delaying treatment protocols, or multiple inoculations, will contribute to a better understanding of the pathogenesis of infection progression to multi-organ dysfunction, systemic inflammatory response, and immune dysregulation (Fig 8). The voluminous data that can be produced by this model may also be useful for identifying new biomarkers, gene expression and cytokine profiles, immune cell populations, and immune cell function. Further examination of the metabolome, microbiome, and other systems will greatly enhance the likelihood of the identification of prognostics, development of diagnostics, and treatment strategies of sepsis and septic shock helping to bring badly needed precision medicine approaches to a poorly misunderstood disease that continues to confound progress in its treatment. To summarize, the combination of this model's longitudinal observation of temporal events, its breadth and magnitude of collected physiologic, metabolic, and organ specific data and most importantly, its clinical faithfulness, is what sets this model apart from murine or other large animal models of sepsis. The uniqueness of this model lies in its attempt to reproduce a complicated series of events leading to a recognizable clinical disease state in reproducible fashion within a realistic timeframe. This model will add to and enrich currently available models if utilized as a platform for further study of diagnostics, markers and therapeutics related to severe infection and sepsis.

Despite the value of our model, we acknowledge a number of limitations. First, in our study, we did not compare the results of our experimental group to an independent negative control group (no inoculation). Instead, we used each animal as its own control by comparing baseline parameters (pre-inoculum) to changes over time (post-inoculum). This design was employed because the primary aim of our work was to follow the progression of events and collect repeated measures data while observing and studying within-subject effects (compare baseline data to after inoculation). We acknowledge that the study of the progression of the systemic responses to infection in a large animal model necessitates prolonged anesthesia and mechanical ventilation and that these interventions, regardless of how well controlled, likely influence a number of the parameters. To minimize the impact of these factors, we took great care to be as consistent as possible in the execution of our detailed experimental protocol. This included participation of the same team members for all experiments and the use of standard operating procedures for preparation and injection of the inoculum and sample collection and processing. Notably, the magnitude of change of some of the hemodynamic and biochemical measurements (e.g., temperature, complete blood count, kidney function) cannot be explained in the absence of a direct effect of bacterial inoculation and are consistent with those that would be expected in a severe, untreated infection in humans. Finally, we recognize that the laparotomy used to visualize the kidney for direct inoculation could incite a surgical inflammatory response and peritoneal or retroperitoneal bacterial seeding could introduce variance into our measurements. We think the likelihood of this is negligible because of the careful procedure we used to inject the bacterial inoculum directly into the kidney. These limitations notwithstanding, our model advances the field of experimental sepsis because it establishes the much needed temporal course of the early systemic responses to severe infection.

## Conclusion

A swine model of untreated infection that employs a renal injection of uropathogenic *E. coli* generates physiologic and biochemical data that are highly relevant to the study of human sepsis. This sets a road map for future assessment of therapeutic, diagnostic, and prognostic modalities while furthering the broader understanding of the pathophysiologic and mechanistic underpinnings of sepsis and septic shock. Thus, this novel model will enable the testing and

validation of critically needed novel therapeutics against systemic infection that can often lead to sepsis and septic shock.

## Supporting information

**S1 Fig. Metabolic changes associated with the natural history of systemic infection.** The adenylate energy charge (**A**) did not change, but the ATP/ADP ratio (**B**) trended downward. Lactate, as measured by NMR (**C**), followed the same trend as the clinical measurement and was not significantly changed during the course of the experiment (ANOVA false discovery rate [FDR] corrected p-value = 0.3). In addition to the nine metabolites with FDR < 10%, there were an additional seven metabolites with FDR of < 15%. These were (**D**) ornithine (FDR = 13%), (**E**) threonine (FDR = 13%), (**F**) AMP (FDR = 13%), (**G**) alanine (FDR = 13%), (**H**) serine (FDR = 13%), (**I**) glutathione (FDR = 15%), and (**J**) creatinine (FDR = 15%). Data are the mean (SE) from four animals. BL = baseline.
(TIF)

**S2 Fig. Metscape generated network of the metabolites of progressive systemic infection.** Of the nine metabolites with an FDR-corrected ANOVA p value of < 0.10, six (inosine mono-phosphate, glutamine, methionine, tyrosine, phenylalanine and histidine) mapped into a single network. The figure was generated by uploading the Kyoto Encyclopedia of Genes and Genomes (KEGG) identification (ID) numbers of the nine metabolites into Metscape (3.1.3 metscape.ncibi.org/), a plugin application for Cytoscape (3.8.0 cytoscape.org). The human library was used. Metabolites are designated by red hexagons with dark red being those that were manually entered into Metscape. The gray squares represent reactions, green round-corner squares, enzymes and blue circles, genes.
(TIF)

**S3 Fig. Carnitine and acetylcarnitine are not detectable early in the course of infection.** Concentrations of (**A**) carnitine were not detectable in any samples at baseline (BL), only one sample had a detectable concentration at 6h but carnitine was detected in all samples by the end of the experiment. Acetylcarnitine (**B**) was only detectable on one sample at BL and 6h and only two samples at the end of the experiment. These metabolites were removed from the primary analysis because of data missingness but are shown here because they have previously been shown to be important as indicators of sepsis severity (see online supplement text). Data are the mean (SE, when applicable) from four animals.
(TIF)

**S4 Fig. Kaplan-Meier survival plot for experimental end times.** Plot of survival times for each animal used in the study.
(TIF)

**S1 Table. Baseline and end of experiment parameters for all animals.** Hemodynamic and biochemical parameters were measured in each animal at baseline (pre-infection) and at the termination of the experiment. These included core temperature, heart rate, mean arterial pressure, central venous pressure (CVP); pulmonary artery pressure (PAP); creatinine, blood urea nitrogen (BUN); glucose, lactate, total bilirubin, alanine transaminase (ALT); platelets, hemoglobin, hematocrit, white blood cells (WBC; with percent neutrophils and lymphocytes); arterial pH; bicarbonate ($HCO_3^-$); end tidal carbon dioxide pressure ($PetCO_2$); oxygen consumption, partial pressure of oxygen ($PaO_2$); fraction of inspired oxygen ($FiO_2$); and central venous oxygen saturation ($ScvO_2$).
(DOCX)

**S2 Table. $^1$H-Nuclear Magnetic Resonance (NMR)-detected and quantified swine whole blood metabolites with Kyoto Encyclopedia of Genes and Genome (KEGG) Identifications (ID).**
(DOCX)

**S1 File. A comprehensive assessment of multi-system responses to a renal inoculation of uropathogenic *E. coli* in swine.**
(DOCX)

**S2 File. Sepsis-manuscript compiled-data.**
(XLSX)

## Acknowledgments

The authors would like to thank Christopher Fry and the staff of Michigan Center for Integrative Research in Critical Care (MCIRCC) for their technical support. We would also like to thank Cora McHugh for her assistance with the acquisition of the metabolomics data.

## Author Contributions

**Conceptualization:** Mohamad Hakam Tiba, Brendan M. McCracken, Robert P. Dickson, Jean A. Nemzek, Rodney C. Daniels, J. Scott VanEpps, Kathleen A. Stringer, Kevin R. Ward.

**Data curation:** Mohamad Hakam Tiba, Brendan M. McCracken, Robert P. Dickson, Jean A. Nemzek, Carmen I. Colmenero, Danielle C. Leander, Thomas L. Flott, Rodney C. Daniels, J. Scott VanEpps, Kathleen A. Stringer.

**Formal analysis:** Mohamad Hakam Tiba, Brendan M. McCracken, Danielle C. Leander, Thomas L. Flott, Kathleen A. Stringer.

**Funding acquisition:** Kathleen A. Stringer.

**Investigation:** Mohamad Hakam Tiba, Brendan M. McCracken, Robert P. Dickson, Jean A. Nemzek, Carmen I. Colmenero, Danielle C. Leander, Thomas L. Flott, Rodney C. Daniels, Kristine E. Konopka, J. Scott VanEpps, Kathleen A. Stringer.

**Methodology:** Mohamad Hakam Tiba, Brendan M. McCracken, Robert P. Dickson, Jean A. Nemzek, J. Scott VanEpps, Kathleen A. Stringer, Kevin R. Ward.

**Project administration:** Mohamad Hakam Tiba, Brendan M. McCracken, Carmen I. Colmenero, Kathleen A. Stringer.

**Resources:** Mohamad Hakam Tiba, Brendan M. McCracken, Robert P. Dickson, Jean A. Nemzek, Rodney C. Daniels, J. Scott VanEpps, Kathleen A. Stringer, Kevin R. Ward.

**Supervision:** Mohamad Hakam Tiba, Brendan M. McCracken, Jean A. Nemzek, Kathleen A. Stringer, Kevin R. Ward.

**Validation:** Mohamad Hakam Tiba, Brendan M. McCracken, Danielle C. Leander, Thomas L. Flott, Rodney C. Daniels, J. Scott VanEpps, Kathleen A. Stringer, Kevin R. Ward.

**Visualization:** Mohamad Hakam Tiba, Brendan M. McCracken, Danielle C. Leander, Thomas L. Flott, Kathleen A. Stringer.

**Writing – original draft:** Mohamad Hakam Tiba, Brendan M. McCracken, Thomas L. Flott, Kathleen A. Stringer.

**Writing – review & editing:** Mohamad Hakam Tiba, Brendan M. McCracken, Robert P. Dickson, Jean A. Nemzek, Carmen I. Colmenero, Danielle C. Leander, Thomas L. Flott, Rodney C. Daniels, Kristine E. Konopka, J. Scott VanEpps, Kathleen A. Stringer, Kevin R. Ward.

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
