## [Decision Letter · Decision Letter 0]

30 Jul 2020

PONE-D-20-18461

Natural history of the systemic responses to a renal inoculation of uropathogenic E. coli in swine

PLOS ONE

Dear Dr. Tiba,

Thank you for submitting your manuscript to PLOS ONE. After careful consideration, we feel that it has merit but does not fully meet PLOS ONE’s publication criteria as it currently stands. Therefore, we invite you to submit a revised version of the manuscript that addresses the points raised during the review process.

We look forward to receiving your revised manuscript.

Kind regards,

Anasuya Sarkar

Academic Editor

PLOS ONE

Journal Requirements:

2.  Please state where the swine used in this study were obtained.

Please also provide survival curves as a supplementary file.

Also, please state the total length of time the experiment was run.

Finally, please state any special training in animal care or handling provided for research staff.

Reviewers' comments:

Reviewer's Responses to Questions

**Comments to the Author**

1. Is the manuscript technically sound, and do the data support the conclusions?

Reviewer #1: Partly

2. Has the statistical analysis been performed appropriately and rigorously? 

Reviewer #1: Yes

3. Have the authors made all data underlying the findings in their manuscript fully available?

Reviewer #1: Yes

4. Is the manuscript presented in an intelligible fashion and written in standard English?

Reviewer #1: Yes

5. Review Comments to the Author

Reviewer #1: Comments to the Authors:

General Comments:

Tiba et al. present a careful summary of the hemodynamic, immunologic and metabolic changes induced after injecting E. coli organisms into the renal pelvis of 5 intubated and anaesthetized pigs. Detailed kinetics over a 9-13 hour period are displayed including cytokine profiles, Hgb, blood cell counts, ETCO2, amino acid levels and essential metabolic measurements. The study provides a strong argument to support the need for better animal models (than murine models) to study sepsis and propose that their model provides a representative pattern of physiological changes that could make a better sepsis model than the murine model.

I have several issues with the work. First of all there is no control animal for the E coli challenge. That is, we cannot distinguish which of the changes seen are due to the infection and which are due to the 12 h instrumentation and anesthesia. This is particularly critical for the metabolic changes that are noted which may well relate to the effects of immobility and fasting as much as from sepsis.

Secondly, the major point is that this is a novel model. However, the paper provides no supportive information to explain why this “novel” model is more representative of sepsis than previously published studies using pigs. Neither does the work necessarily demonstrate that this is a better model. There is no comparison model like cecal ligation and puncture or i.p. injection of E coli.

Specific Comments:

1 Details in the description of the model that injects live E coli into the renal pelvis are lacking. This is the key uniqueness of the model yet hard to tell what was actually done here. The methods mention a 22 gauge catheter but don’t’ mention how this was done. Was it percutaneous under ultrasound? There is a mention in the discussion of the study limitations that argues that the “need for a laparoscopy to visualize the kidney could introduce variance into the measurements”. This is the first mention that I found of the model. If this was done for all 5 animals, there is concern that the laparoscopy itself may change the model dramatically by providing a peritoneal or retroperitoneal site for extra renal seeding of organisms. Was the wound closed or left open?

2 Line 353 mentions that one of the animals showed an early burst of serum TNF at about 90 min. The attribution to surgical procedures is possible but it should be noted that human studies have classically noted an early peak of TNF 90 min. after endotoxin infusions.

3 What was the timing of the surgical procedures and was laparotomy the main surgical procedure or does this refer to line placements and intubation?

4 This sepsis model is provided as a better “mouse trap” so to speak for sepsis work that would be more human-like. It would be helpful to show in a Table or Discussion format exactly the ways that this model improves upon what has already been published regarding animal models, to include cost, reproducibility, aspects that mimic the human condition, ability to develop biomarkers etc.

5 In this context, Figure 8 presents a typical model of the sepsis events. Might be better to use this space to provide a diagram or table that describes the summary advantages of this model over murine models.

Minor comments:

The title “Natural history of ….” seems odd word choice for a 12 hour study.

6. PLOS authors have the option to publish the peer review history of their article (what does this mean?). If published, this will include your full peer review and any attached files.

Reviewer #1: No

---

## [Author Response · Author response to Decision Letter 0]

1 Sep 2020

Journal Requirements:

1. Please ensure that your manuscript meets PLOS ONE's style requirements, including those for file naming. The PLOS ONE style templates can be found at https://journals.plos.org/plosone/s/file?id=wjVg/PLOSOne_formatting_sample_main_body.pdf

And

The manuscript and files names have been formatted to meet PLoS One style requirements

2. Please state where the swine used in this study were obtained.

Response: All animals used in this study were procured from the Michigan State University Swine Teaching & Research Center. The manuscript has been updated to reflect this. (redlined manuscript: lines 67-68)

3. Please also provide survival curves as a supplementary file.

Response: The survival curve has been submitted as supplementary file “S4 Fig”.

4. Also, please state the total length of time the experiment was run

Response: The total length of the experiment was designed to run for up to 24 hours from inoculation. The manuscript has been updated and this information is now included (redlined manuscript: lines 111 and 125). 

5. Finally, please state any special training in animal care or handling provided for research staff.

Response: All staff participating in these experiments received specialized training and have completed the following courses from the University of Michigan’s Unit for Laboratory Animal Medicine:

-ULAM-C10000 Orientation to Animal Care and Use at the University of Michigan

-ULAM-C11400 Introduction to the Laboratory Swine

-ULAM-C11771 Non-Rodent Mammalian Survival Surgery

The Manuscript has been updated with this information (redlined manuscript: lines 68-70).

Reviewers' comments:

Reviewer's Responses to Questions

1. Is the manuscript technically sound, and do the data support the conclusions?

‘The manuscript must describe a technically sound piece of scientific research with data that supports the conclusions. Experiments must have been conducted rigorously, with appropriate controls, replication, and sample sizes. The conclusions must be drawn appropriately based on the data presented.” 

 Reviewer #1: Partly

2. Has the statistical analysis been performed appropriately and rigorously? 

Reviewer #1: Yes

3. Have the authors made all data underlying the findings in their manuscript fully available?

“The PLOS Data policy requires authors to make all data underlying the findings described in their manuscript fully available without restriction, with rare exception (please refer to the Data Availability Statement in the manuscript PDF file). The data should be provided as part of the manuscript or its supporting information, or deposited to a public repository. For example, in addition to summary statistics, the data points behind means, medians and variance measures should be available. If there are restrictions on publicly sharing data—e.g. participant privacy or use of data from a third party—those must be specified.”

Reviewer #1: Yes

4. Is the manuscript presented in an intelligible fashion and written in standard English?

“PLOS ONE does not copyedit accepted manuscripts, so the language in submitted articles must be clear, correct, and unambiguous. Any typographical or grammatical errors should be corrected at revision, so please note any specific errors here.”

Reviewer #1: Yes

Review Comments to the Author

Reviewer #1: Comments to the Authors:

General Comments: Tiba et al. present a careful summary of the hemodynamic, immunologic and metabolic changes induced after injecting E. coli organisms into the renal pelvis of 5 intubated and anaesthetized pigs. Detailed kinetics over a 9-13-hour period are displayed including cytokine profiles, Hgb, blood cell counts, ETCO2, amino acid levels and essential metabolic measurements. The study provides a strong argument to support the need for better animal models (than murine models) to study sepsis and propose that their model provides a representative pattern of physiological changes that could make a better sepsis model than the murine model.

I have several issues with the work.

Comment 1: 

First of all, there is no control animal for the E coli challenge. That is, we cannot distinguish which of the changes seen are due to the infection and which are due to the 12 h instrumentation and anesthesia. This is particularly critical for the metabolic changes that are noted which may well relate to the effects of immobility and fasting as much as from sepsis.

Response: We thank the reviewer for this comment and we certainly appreciate this concern. Although we haven’t compared the results of our experimental group to control (no inoculation), the absence of a control group may be justified for this type of investigation. In this pilot study, we standardized all anesthetic and surgical manipulation for all five animals. This was done before inoculation and was then longitudinally tracked. In addition, we contend that in this type of observational study, for which the E. coli inoculation was the only experimental manipulation, a comparison to control is not necessary since the primary aim of the work was to follow the progression of events. Furthermore, we collected all data in a repeated measures fashion while observing and studying within-subject effects. This allowed us to compare baseline data (prior to inoculation) to those after inoculation. 

Although some of the observed effects may be attributable to surgical manipulation and prolonged anesthesia, the magnitude of change in many of the hemodynamic and biochemical measurements (e.g., temperature, CBC, kidney function) cannot simply be explained by these factors. It is reasonable to conclude that these changes are due to the direct effect of bacterial inoculation like that which may be observed in a severe untreated infection in humans which, of course, is not ethically feasible. We acknowledged this point as a limitation in our original submission with a description of the mitigation effort to minimize such effects. We have expanded the details about this limitation by adding “…we used each animal as its own control by comparing baseline parameters (pre-inoculum) to changes over time (post-inoculum). This design was employed because the primary aim of our work was to follow the progression of events and collect repeated measures data while observing and studying within-subject effects (compare baseline data to after inoculation).” to the discussion section. We have also altered the title of the paper to remove the implication that the time course is a “natural history”. 

The manuscript has been updated to address the reviewer’s comment (redlined manuscript: lines 420-428). 

Comment 2: 

Secondly, the major point is that this is a novel model. However, the paper provides no supportive information to explain why this “novel” model is more representative of sepsis than previously published studies using pigs. Neither does the work necessarily demonstrate that this is a better model. There is no comparison model like cecal ligation and puncture or i.p. injection of E coli.

Response: We thank the reviewer for this comment. The novelty of our model lies in its faithfulness to the clinical condition of sepsis. It describes the systemic events that occur during the manifestation of infection. This starts with the knowledge that pyelonephritis can progress to one of the leading causes of sepsis, urosepsis (second only to pneumonia); a cause of sepsis that is far more clinically common than bacterial peritonitis (the infection which is mimicked by cecal ligation and puncture or i.p. injection of bacteria). The model’s longitudinal observation of temporal events, its breadth and magnitude of physiologic, metabolic, and organ specific data collected, sets this model apart from murine or other large animal models of sepsis. The uniqueness of this model also lies in its utility for the identification of a complicated series of events that leads to a recognizable clinical disease state in reproducible fashion within a realistic timeframe. This model is envisioned to provide a more accessible platform for further study of diagnostics, markers and therapeutics related to severe infection and sepsis. 

The manuscript has been updated to address the reviewer’s comment (redlined manuscript: lines 413-419)

Specific Comments:

Comment 3: 

Details in the description of the model that injects live E coli into the renal pelvis are lacking. This is the key uniqueness of the model yet hard to tell what was actually done here. The methods mention a 22-gauge catheter but don’t’ mention how this was done. Was it percutaneous under ultrasound? There is a mention in the discussion of the study limitations that argues that the “need for a laparoscopy to visualize the kidney could introduce variance into the measurements”. This is the first mention that I found of the model. If this was done for all 5 animals, there is concern that the laparoscopy itself may change the model dramatically by providing a peritoneal or retroperitoneal site for extra renal seeding of organisms. Was the wound closed or left open?

Response: A 22-gauge catheter was placed in the kidney under direct visualization through the laparotomy. The seeding of bacterial organisms outside of the kidney was minimized by maintaining a slow and constant infusion rate (0.3 mL/min), and proper placement of the catheter, during the infusion. We also applied direct pressure using gauze to the infusion site of the kidney. As such, the primary inflammatory response is largely driven by the direct renal inoculation and less likely due to peritonitis. The laparotomy was closed immediately following the completion of the infusion, leaving the Foley catheter and Potts loop ureter occlusion externalized. 

These details have been added to the methods section (redlined manuscript: lines 101-106) and limitations of this procedure has been addressed in the discussion (redlined manuscript: lines 431-439). 

Comment 4:

Line 353 mentions that one of the animals showed an early burst of serum TNF at about 90 min. The attribution to surgical procedures is possible but it should be noted that human studies have classically noted an early peak of TNF 90 min. after endotoxin infusions.

Response: In this statement, we intended to suggest the burst of TNF-α immediately following bacterial infusion was more likely attributable to the host’s inflammatory response to the pathogen, and less likely related to the surgical procedures, consistent with the reviewer’s mention of previous studies. The manuscript has been updated to clarify this point. Specifically, we have added the statement: “Of note, as shown in Fig 6, there was an early post-inoculum rise in concentrations of TNF-α and subsequently IL-6, that appears to occur independent of the invasive surgical procedures. This suggests that such increases are primarily due to the host’s response to the newly introduced bacteria into the kidney.” 

These details have been added to the discussion section (redlined manuscript: lines 371-374).

Comment 5: 

What was the timing of the surgical procedures and was laparotomy the main surgical procedure or does this refer to line placements and intubation?

Response: Surgical procedures consisted of the insertion of catheters into surgically exposed vessels (i.e., arterial and venous cutdown), laparotomy, and Foley catheter placement. All procedures were carried out concurrently within 1.5 to 2 hours.

These details have been added to the methods section (redlined manuscript: lines 86-89). 

Comment 6: 

This sepsis model is provided as a better “mouse trap” so to speak for sepsis work that would be more human-like. It would be helpful to show in a Table or Discussion format exactly the ways that this model improves upon what has already been published regarding animal models, to include cost, reproducibility, aspects that mimic the human condition, ability to develop biomarkers etc.

Response: We thank the reviewer for this comment. We appreciate that a table would be helpful but we think it is much more efficient to address these elements in the discussion (and the introduction which sets the premise for the work). However, in doing so, we view that an extensive comparison of published experimental models and their advantages and disadvantages is beyond the scope of the paper. To generate this type of review would, in our view, require head-to-head comparisons across models as well as procurement of information that may not be readily or reliably available in the literature (e.g., costs). Of note, there have been numerous commentaries about the utility and translatability of experimental sepsis models to the human condition. So much concern has been raised about the translational gap in sepsis that it prompted the National Institute of General Medical Sciences of the U.S. National Institutes of Health to make rodent-based sepsis studies a low priority (https://grants.nih.gov/grants/guide/notice-files/NOT-GM-19-054.html). 

Comment 7:

In this context, Figure 8 presents a typical model of the sepsis events. Might be better to use this space to provide a diagram or table that describes the summary advantages of this model over murine models.

Response: We would very much like to keep Figure 8 (please also see our response to Comment 6). It represents an overview of the timeline of the events of our model and some of its unique aspects. In our opinion, this provides a visual and important summation of the model for readers. We have updated it to make some aspects of it more clear. 

Minor comments:

Comment 8:

The title “Natural history of ….” seems odd word choice for a 12-hour study.

Response: The title of the manuscript has been changed to “A Comprehensive Assessment of Multi-System Responses to a Renal Inoculation of Uropathogenic E. coli in Swine.”

---

## [Decision Letter · Decision Letter 1]

12 Oct 2020

PONE-D-20-18461R1

A comprehensive assessment of multi-system responses to a renal inoculation of uropathogenic E. coli in swine

PLOS ONE

Dear Dr. Tiba,

Thank you for submitting your manuscript to PLOS ONE. After careful consideration, we feel that it has merit but does not fully meet PLOS ONE’s publication criteria as it currently stands. Therefore, we invite you to submit a revised version of the manuscript that addresses the points raised during the review process.

We look forward to receiving your revised manuscript.

Kind regards,

Anasuya Sarkar

Academic Editor

PLOS ONE

Reviewers' comments:

Reviewer's Responses to Questions

**Comments to the Author**

1. If the authors have adequately addressed your comments raised in a previous round of review and you feel that this manuscript is now acceptable for publication, you may indicate that here to bypass the “Comments to the Author” section, enter your conflict of interest statement in the “Confidential to Editor” section, and submit your "Accept" recommendation.

Reviewer #1: (No Response)

2. Is the manuscript technically sound, and do the data support the conclusions?

Reviewer #1: Yes

3. Has the statistical analysis been performed appropriately and rigorously? 

Reviewer #1: Yes

4. Have the authors made all data underlying the findings in their manuscript fully available?

Reviewer #1: Yes

5. Is the manuscript presented in an intelligible fashion and written in standard English?

Reviewer #1: Yes

6. Review Comments to the Author

Reviewer #1: The manuscript has been improved and the changes have largely answered the concerns that were expressed. However, the suggestion that this approach is novel deserves a supportive discussion. A paragraph in the discussion based upon what is already known about porcine models is needed to put the current work into context.

It is necessary that the manuscript make it explicit why this models is more representative of sepsis than previously published studies using pigs. Perhaps taking advantage of the reviews that the author as mentioned, the discussion could add a paragraph to put the work into better context

7. PLOS authors have the option to publish the peer review history of their article (what does this mean?). If published, this will include your full peer review and any attached files.

Reviewer #1: No

---

## [Author Response · Author response to Decision Letter 1]

28 Oct 2020

Comments to the Author

1. If the authors have adequately addressed your comments raised in a previous round of review and you feel that this manuscript is now acceptable for publication, you may indicate that here to bypass the “Comments to the Author” section, enter your conflict of interest statement in the “Confidential to Editor” section, and submit your "Accept" recommendation.

 Reviewer #1: (No Response)

2. Is the manuscript technically sound, and do the data support the conclusions?

Reviewer #1: Yes

3. Has the statistical analysis been performed appropriately and rigorously? 

Reviewer #1: Yes

4. Have the authors made all data underlying the findings in their manuscript fully available?

Reviewer #1: Yes

5. Is the manuscript presented in an intelligible fashion and written in standard English?

Reviewer #1: Yes

6. Review Comments to the Author

Reviewer #1: 

Comment: The manuscript has been improved and the changes have largely answered the concerns that were expressed. However, the suggestion that this approach is novel deserves a supportive discussion. A paragraph in the discussion based upon what is already known about porcine models is needed to put the current work into context.

It is necessary that the manuscript make it explicit why this models is more representative of sepsis than previously published studies using pigs. Perhaps taking advantage of the reviews that the author as mentioned, the discussion could add a paragraph to put the work into better context

Response: We thank the reviewer for this comment. We have added a paragraph to highlight the novelty of our model and what sets it apart from other models. In addition to our model’s faithfulness to the clinical course of severe infection, the model’s longitudinal observation of temporal events, its breadth and magnitude of physiologic, metabolic, and organ specific data collected, sets it apart from murine or other large animal models of sepsis. These features make our model distinct from other porcine models of sepsis particularly since, to date, most have induced sepsis via peritonitis or by direct intravenous infusion of either a pathogen or endotoxin. Furthermore, many have employed interventions such as fluid supplementation, antibiotics or vasopressors which mask the natural course of illness. We also added citation [47] which lists the MQTiPPS review Part II recommendations. Within other recommendations, it advocates for the discontinuation of endotoxin challenge use as well as modeling of the sepsis syndromes at sites other than the peritoneal cavity (e.g., lung, urinary tract, brain). Our model aligns closely with these recommendations. 

The manuscript has been updated to address the reviewer’s comment and few other formatting issues (redlined manuscript: lines 423-426, 432-437, 671-679, and 685-687).

---

## [Editor Report · Decision Letter 2]

24 Nov 2020

A comprehensive assessment of multi-system responses to a renal inoculation of uropathogenic E. coli in swine

PONE-D-20-18461R2

Dear Dr. Tiba,

We’re pleased to inform you that your manuscript has been judged scientifically suitable for publication and will be formally accepted for publication once it meets all outstanding technical requirements.

Kind regards,

Anasuya Sarkar

Academic Editor

PLOS ONE
---

## [Editor Report · Acceptance letter]

1 Dec 2020

PONE-D-20-18461R2 

A comprehensive assessment of multi-system responses to a renal inoculation of uropathogenic *E. coli* in swine 

Dear Dr. Tiba:

I'm pleased to inform you that your manuscript has been deemed suitable for publication in PLOS ONE. Congratulations! Your manuscript is now with our production department. 

Kind regards, 

on behalf of

Dr. Anasuya Sarkar 

Academic Editor

PLOS ONE